# Variability in digestive and respiratory tract *Ace2* expression is associated with the microbiome

**Sean T. Koester**[1☯], **Naisi Li**[1☯], **Daniel M. Lachance**[1,2], **Norma M. Morella**[1], **Neelendu Dey**[1,3,4]*

1 Clinical Research Division, Fred Hutchinson Cancer Research Center, Seattle, WA, United States of America, 2 Molecular Engineering & Sciences Institute, University of Washington, Seattle, WA, United States of America, 3 Microbiome Research Initiative, Fred Hutchinson Cancer Research Center, Seattle, WA, United States of America, 4 Division of Gastroenterology, Department of Medicine, University of Washington, Seattle, WA, United States of America

☯ These authors contributed equally to this work.

* ndey@fredhutch.org

**Data Availability Statement:** All relevant data are within the paper and its Supporting Information files.

**Funding:** This work was supported by grants from the NIH (NIDDK K08 DK111941 and NCI Cancer Center Support Grant P30 CA015704 to ND);

## Abstract

COVID-19 (coronavirus disease 2019) patients exhibiting gastrointestinal symptoms are reported to have worse prognosis. *Ace2* (angiotensin-converting enzyme 2), the gene encoding the host protein to which SARS-CoV-2 spike proteins bind, is expressed in the gut and therefore may be a target for preventing or reducing severity of COVID-19. Here we test the hypothesis that *Ace2* expression in the gastrointestinal and respiratory tracts is modulated by the microbiome. We used quantitative PCR to profile *Ace2* expression in germ-free mice, conventional raised specific pathogen-free mice, and gnotobiotic mice colonized with different microbiota. Intestinal *Ace2* expression levels were significantly higher in germ-free mice compared to conventional mice. A similar trend was observed in the respiratory tract. Intriguingly, microbiota depletion via antibiotics partially recapitulated the germ-free phenotype, suggesting potential for microbiome-mediated regulation of *Ace2* expression. Variability in intestinal *Ace2* expression was observed in gnotobiotic mice colonized with different microbiota, partially attributable to differences in microbiome-encoded proteases and peptidases. Together, these data suggest that the microbiome may be one modifiable factor determining COVID-19 infection risk and disease severity.

## Introduction

Epidemiologic studies have reported variable rates of gastrointestinal symptoms among COVID-19 (coronavirus disease 2019) patients [1,2]. This variability may relate to disease outcomes, as gastrointestinal symptoms portend worse prognosis [1]. Although fecal-oral transmission of SARS-CoV-2 as a widespread infectious mechanism is debated, live virus has been recovered from fecal samples, suggesting the gut may serve as a viral reservoir. *Ace2* (angiotensin-converting enzyme 2), the gene encoding the host protein to which SARS-CoV-2 spike

institutional funds from the Fred Hutchinson
Cancer Research Center (Pathogen-Associated
Malignancies Integrated Research Center,
Microbiome Research Initiative, and donor-
sponsored COVID-19 Evergreen pilot study funds
to ND); and postdoctoral fellowship support from
the Washington Research Foundation (NMM).

**Competing interests:** The authors have declared
that no competing interests exist.

proteins bind, is expressed in the gut, where it mediates amino acid transport [3]. ACE2 has
been linked to the gut microbiome [4,5]; by association, the microbiome may be a target for
preventing COVID-19 or mitigating severity thereof.

Here we tested the hypothesis that the gut microbiome modulates *Ace2* expression in the
gastrointestinal and respiratory tracts. We studied germ-free (GF) mice, conventional raised
specific pathogen-free (SPF) mice, and gnotobiotic mice colonized with different microbiota.
Small intestinal and colonic *Ace2* expression levels were significantly higher in GF mice com-
pared to conventional mice; this difference was modest in the respiratory tract. Intriguingly,
antibiotic-mediated microbiota depletion in SPF mice partially rescued the phenotype seen in
GF mice, suggesting a capacity for microbiome-mediated regulation of *Ace2* expression. Vari-
ability in intestinal *Ace2* expression was observed in gnotobiotic mice colonized with different
microbiota, suggesting that population-wide differences in *Ace2* expression may in part be
attributable to differences in structure and function of the gut microbiome. Together, these
data suggest that the microbiome may be one modifiable factor determining COVID-19 dis-
ease severity.

## Materials and methods

### Animal husbandry

Male and female Swiss-Webster and C57BL/6 mice were studied using protocols approved by
the Institutional Animal Care and Use Committees of the University of Washington and Fred
Hutchinson Cancer Research Center. Gnotobiotic mouse tissue samples were harvested as
part of prior experiments unrelated to the present study. The fecal microbiota suspension
(derived from fecal pellets collected from conventionally housed wild-type SPF mice) and bac-
terial consortia used for colonization were prepared in an anaerobic chamber. Defined bacte-
rial consortia were prepared by first anaerobically monoculturing bacterial strains in rich
growth media (as previously described [6]) to mid-log phase (optical density at 600 nm of 0.4),
combining in equal volumes, and then storing in 25% glycerol at -80˚C until use. Mice were
colonized via oral gavage by the same individual (N.L.) in all mice in order to minimize vari-
ability. Fresh fecal pellets were snap-frozen in liquid nitrogen and stored at -80˚C until use.
Vancomycin (1 g/L), metronidazole (1 g/L), and neomycin (0.5 g/L) antibiotics (Millipore
Sigma, St. Louis, MO) were delivered via drinking water containing 2% sucrose (20 g/L) over
10 days.

### Confirmation of microbiota depletion using droplet digital PCR (ddPCR)

Fresh mouse pellets were collected, homogenized via bead-beating in a TissueLyser II (Qiagen,
Hilden, Germany; 3 min at frequency 30/sec; tubes contained 0.1 mm Zirconia beads, one 4
mm steel ball, and buffer), and pelleted via centrifugation. The supernatant was diluted 1:10
and used for 16S rRNA PCR reactions (95˚C for 5 min; then 40 cycles of 95˚C for 30 sec, 60˚C
for 1 min; then 4˚C for 5 min and 90˚C for 5 min) prepared in triplicate using BioRad QX200™
ddPCR™ EvaGreen® SuperMix with primers 926F and 1062R [7]. Negative controls were pre-
pared using water as a template. Positive controls were prepared using 1 ng/μl purified bacte-
rial genomic DNA extracted from *Clostridium scindens*. Droplet preparation and signal
acquisition was performed on the QX200™ droplet reader and analyzed using associated Quan-
taSoft™ software using manufacturer specifications. The threshold for background fluorescence
was set using background signal from negative controls. The signal was then normalized to the
original fecal pellet weights and corrected for initial dilution (**S1 Fig**).

### RNA isolation

Respiratory and intestinal tract tissues were stored in RNA*later* (Thermo Fisher Scientific Inc., Waltham, MA) at 4°C for 24 hours and then -20°C until use. 20 mg of each tissue sample was subjected to bead-beating in a TissueLyser II. RNA was purified using RNeasy Mini kits (Qiagen).

### Quantitative reverse transcription PCR

*Ace2* and *Gapdh* expression levels were quantified using the QuantiNova Probe RT-PCR kit and LNA probes (Qiagen), with input of 80 ng RNA/sample. *Ace2* expression was normalized to *Gapdh* expression using the formula $2^{-\Delta Ct}$, where $\Delta Ct$ is equal to the difference in mean cycle threshold (Ct) values between *Ace2* and *Gapdh*. The relative change in *Ace2* expression compared to a baseline state was calculated using the equation $2^{-\Delta\Delta Ct}$.

### Quantification of bacterial genome-encoded proteases and peptidases

All bacterial species used to assemble synthetic consortia were purchased from their respective vendors (American Type Culture Collection (ATCC) or the German Collection of Microorganisms and Cell Culture (DSM)); draft genomes of these type-strains are publicly available via NCBI. The most-recent genome assemblies of all bacteria in synthetic communities were downloaded from the NCBI Prokaryotic RefSeq database and annotated using Prokka (version 1.14.5) [8]. Gene annotations that included "peptidase" or "protease" were counted, with sums represented in **S2 Table**.

### Estimation of microbiome-encoded proteases and peptidases in a metagenomic dataset

Metagenomic sequencing data (quality-filtered and human-genome-filtered FASTQ files generated from 15 COVID-19 patients, 6 non-COVID-19 pneumonia patients, and 15 healthy controls for a total of 36 fecal samples) were downloaded from the NCBI Sequence Read Archive (accession PRJNA624223). One million reads were randomly selected from each sample and mapped using the DIAMOND aligner (diamond blastx—id 60—max-target-seqs 5—evalue .00001) to a reference database comprising 277,176 bacterial protease and peptidase sequences representing 57 gut bacterial genera that was downloaded from the MEROPS database [9] (https://www.ebi.ac.uk/merops/download_list.shtml). Total reads strongly aligning to proteases and peptidases (60% amino acid identity; E-value $\leq 10^{-5}$) were compared between COVID-19 cases and pneumonia/healthy controls, with statistical significance calculated using the Kruskal-Wallis test.

### Data analysis

Statistical comparisons were performed in R (version 4.0.0). Figures were generated using R using native functions and *ggplot2* (version 3.3.0), and then assembled in Adobe Illustrator.

## Results

### The microbiome is associated with reduced *Ace2* expression

We performed quantitative reverse transcription PCR (RT-qPCR) to assess *Ace2* expression levels in the trachea, left and right lungs, proximal and distal small intestine, and proximal and distal colon harvested from SPF and GF mice ($n$ = 11-15/cohort). The variability in *Ace2* expression that we observed is consistent with a prior study assessing expression in different

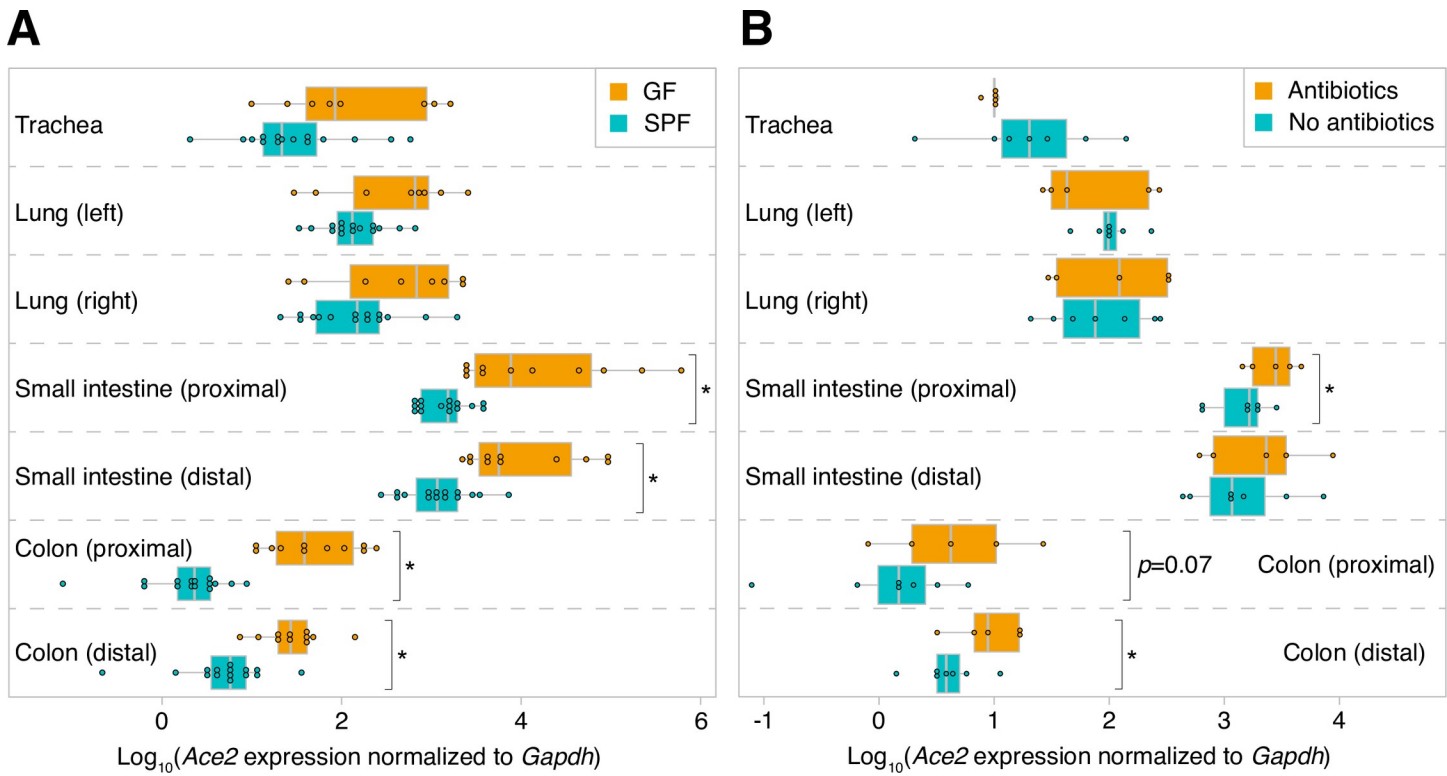

**Fig 1. The microbiome is associated with reduced gastrointestinal and respiratory tract *Ace2* expression. A**. *Ace2* expression in GF and SPF mice. **B**. Antibiotics can modulate intestinal *Ace2* expression. Statistical significance was determined using a two-tailed Student's *t*-test in (A) and one-tailed Student's *t*-test in (B); *, $p < 0.05$.

tissues [10]. Both the microbiome and sample type were significant determinants of *Ace2* expression ($p < 10^{-17}$, $F_1 = 94$ [microbiome] and $p < 10^{-50}$, $F_6 = 98$ [sample type], two-way ANOVA of $\log_{10}$-tranformed normalized *Ace2* expression levels; **S1 Table**). Compared to SPF mice, GF mice had significantly higher levels of *Ace2* throughout the small intestine and colon (**Fig 1A**). Similar but non-significant trends were seen in the trachea and lungs. To our knowledge, there are no prior data linking the microbiome to respiratory tract *Ace2* expression. Intriguingly, SPF mice subjected to microbiota depletion via antibiotic administration for 10 days had higher intestinal *Ace2* expression compared to mice that did not receive antibiotics ($n = 5$-7/cohort, **Fig 1B**). Nonetheless, *Ace2* transcript levels were not as high after antibiotics as in the GF state ($p = 0.08$ and $p = 0.04$ in proximal and distal colon, respectively, two-tailed Student's *t*-test). This partial recapitulation of the GF phenotype in the gut, together with the absence of a measurable effect of antibiotics in the respiratory tract, suggests that antibiotic use may not have sizable immediate effects on host *Ace2* expression.

## Microbiota-dependent variability in intestinal *Ace2* expression

Tissues harvested as part of prior gnotobiotic experiments enabled us to assess whether *Ace2* expression varies in different host and environmental contexts. Gnotobiotic mice colonized with one of two synthetic 6-member communities (**S2 Table**) or with a complete mouse microbiota (via fecal microbiota transplantation using SPF mouse donors) had microbiota-dependent regional variability in gut *Ace2* expression ($p < 0.01$, $F_3 = 5$ [microbiota] and $p < 10^{-21}$, $F_3 = 246$ [sample type], two-way ANOVA of $\log_{10}$-tranformed normalized *Ace2* expression levels), with the greatest effects seen in the distal small intestine and proximal colon (**Fig 2A**).

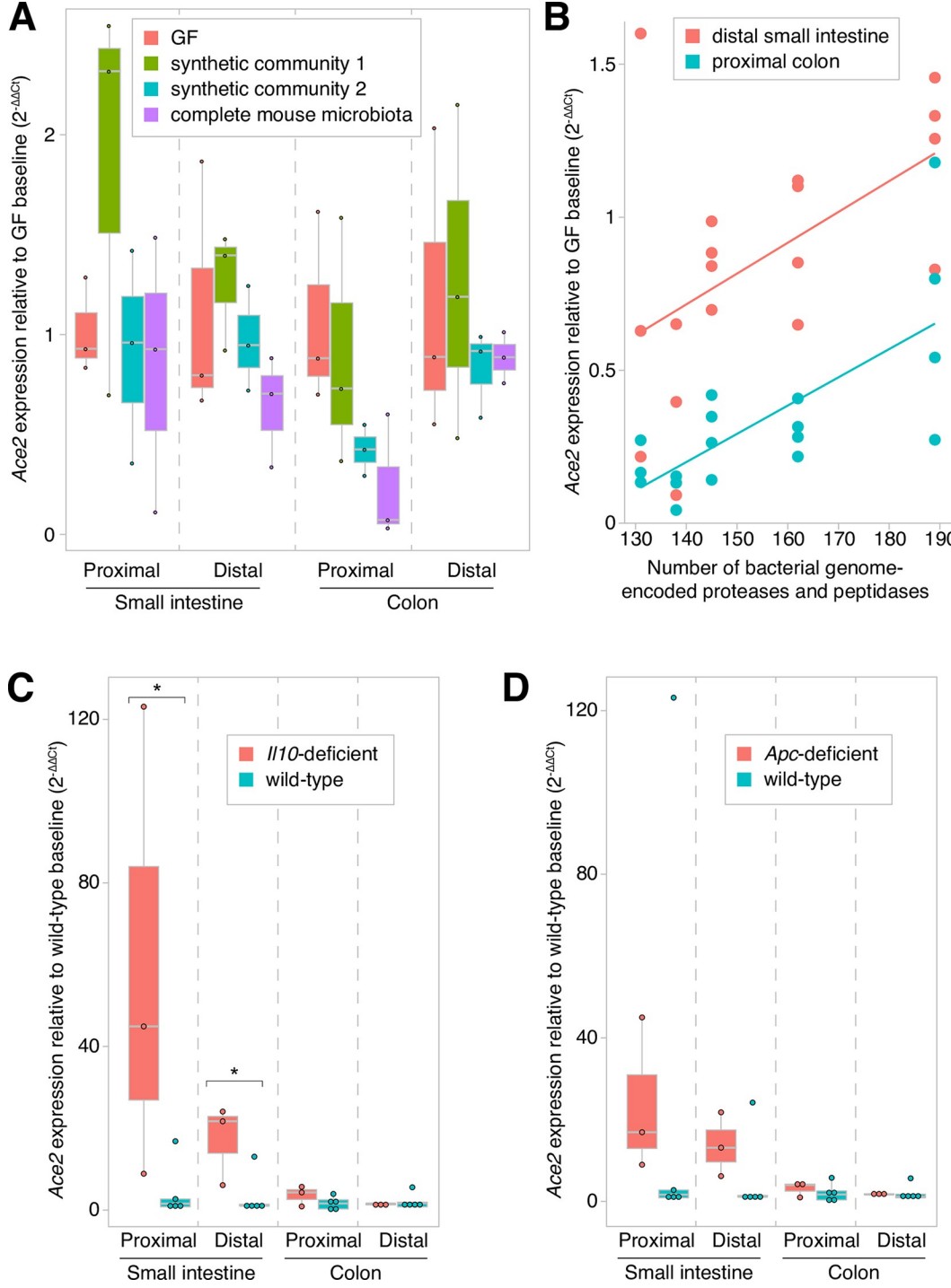

**Fig 2. Variability in intestinal *Ace2* expression in mice.** Gut *Ace2* expression varied in relation to (**A**) different microbiota, (**B**) microbially-encoded proteases and peptidases, (**C**) *Il10*-deficiency, and (**D**) *Apc*-deficiency. Statistical significance was determined using a two-tailed Student's *t*-test; *, $p<0.05$.

Given ACE2's role in intestinal amino acid transport, we postulated that gut microbial proteases and peptidases may explain this observed variation in *Ace2* expression. Consistent with this notion, a prior report demonstrated that dietary supplementation of free amino acids

induced greater *Ace2* expression in the distal small intestine [11]. Therefore, we hypothesized that gut microbial communities with greater proteolytic and peptidase activities would proportionally induce intestinal *Ace2* expression. To test this hypothesis, we quantified *Ace2* expression in the distal small intestine and proximal colon in gnotobiotic mice colonized with one of five different 5-6-member bacterial consortia whose metagenomes encoded between 131 and 189 proteases and peptidases (**S2 Table**). Indeed, we observed a significant correlation between encoded proteases/peptidases and both small intestinal and colonic *Ace2* expression (Spearman correlation *rho* = 0.53, *p* = 0.02 and *rho* = 0.73, *p* = 0.0005, respectively; **Fig 2B**).

We next assessed whether human gut microbiome-encoded proteases/peptidases differ between healthy individuals and COVID-19 patients. We reanalyzed recently published data generated from 15 COVID-19 patients and two sets of controls (6 patients with pneumonia and 15 healthy individuals) [12]. We did not observe a significant relationship between proportions of total reads mapping to proteases/peptidases and disease status (*p*>0.05, Kruskal-Wallis test).

## Host and dietary factors

A mutation in *Il10*, which predisposes to intestinal inflammation in humans and mouse models, was associated with significantly increased small intestinal *Ace2* expression in GF mice (*p* = 0.005, Student's two-tailed *t*-test; **Fig 2C**). These findings are consistent with a study that found greater *Ace2* tissue concentrations in patients with inflammatory bowel diseases [13]. Several variables were not significant determinants of gut or lung *Ace2* expression: genetic background (C57BL/6 versus Swiss-Webster mice); *Ret* (gene critical to enteric nervous system development); *Apc* (tumor suppressor commonly mutated in colorectal cancer; **Fig 2D**); gender; age (comparing 9-16-week-old SPF wild-type mice); or a high-fat diet (*p*>0.05, one-way ANOVA tests performed for each sample tissue type for each variable; specific comparisons denoted in **S1 Table**).

## Discussion

Here we report an association between the gut microbiome and *Ace2* expression in the respiratory and gastrointestinal tracts, and a correlation between gut microbiome-encoded proteases/peptidases and intestinal *Ace2* expression in mice. Although it is unclear whether the observed effect size of the microbiome is clinically meaningful, it is comparable to the effect size of cigarette smoking [14,15], a risk factor for more severe disease among individuals with COVID-19. Thus, the microbiome could theoretically also impact COVID-19 severity.

Our results suggest that gut microbial protein digestion and amino acid liberation may modulate intestinal *Ace2* expression. However, our findings do not explain why GF mice, which have no bacterial proteases/peptidases, have higher *Ace2* expression. One possible explanation is cytokines regulate the expression of *Ace2*. Higher intestinal *Ace2* mRNA levels among *Il10* mutant mice supports this hypothesis. In this scenario, *Ace2* expression is an indirect effect of microbial depletion. Other mechanisms may explain this observation, such as lower expression of peptidase inhibitors in GF mice [16].

In an analysis of fecal metagenomic data generated from a recent study of COVID-19 patients and controls, we found that abundances of microbiome-encoded proteases/peptidases were not significantly different. This finding could potentially be confounded by variation in diet and regulators of gene expression, and our findings are limited by the small study size. In addition, fecal sampling *prior to* SARS-CoV-2 infection would be more informative in assessing the gut microbiome's relationship to COVID-19 susceptibility, as *Ace2* expression can change after infection. However, a caveat to this analysis is that core metabolic pathways are

relatively evenly encoded in the human gut microbiome [17]; therefore, any variation in *expressed* proteases and peptidases that might influence host *Ace2* expression is not captured in our analysis.

Additional studies are warranted to further delineate mechanisms through which the microbiome regulates *Ace2* expression in the respiratory and gastrointestinal tracts. A recent study identified transcriptional factors regulating *Ace2* expression in the gut [18], including Gata4, which is known to be regulated by the microbiota [19]. The partial recapitulation of the GF phenotype that we observed with antibiotics further raises the prospect of differential temporal dynamics of the relationships between the microbiome and the lung versus gut with respect to *Ace2* expression. Perhaps early-life or longer-term microbial exposures are necessary for modulating *Ace2* expression in the lung. Understanding the kinetics and magnitudes of these effects may have implications for antibiotic (and probiotic) use. Protein source and diet composition may be risk-modifying variables. Understanding these links could explain variability in COVID-19 severity and motivate strategies for modulating *Ace2* in order to decrease susceptibility to infection (e.g. upregulating *Ace2* in the elderly, in whom it is suspected that ACE2 levels become dangerously low [20]).

## Supporting information

**S1 Fig. Microbiota depletion verification through ddPCR.** Copies of 16S rRNA genes per μl of homogenate per mg of feces (means of the three technical replicates ± standard deviations; each dot represents a single mouse; values plotted along y-axis in log scale). Positive and negative controls similarly represent means (thick horizontal lines) ± standard deviations (pair of thin horizontal lines above and below each thick line).
(TIF)

**S1 Table. *Ace2* expression levels and associated metadata.**
(XLSX)

**S2 Table. Microbiota used in mouse studies.**
(XLSX)

## Author Contributions

**Conceptualization:** Neelendu Dey.

**Data curation:** Sean T. Koester, Naisi Li, Neelendu Dey.

**Formal analysis:** Sean T. Koester, Daniel M. Lachance, Norma M. Morella, Neelendu Dey.

**Funding acquisition:** Neelendu Dey.

**Investigation:** Sean T. Koester, Naisi Li, Norma M. Morella, Neelendu Dey.

**Methodology:** Sean T. Koester, Naisi Li, Norma M. Morella, Neelendu Dey.

**Project administration:** Neelendu Dey.

**Resources:** Neelendu Dey.

**Software:** Sean T. Koester, Daniel M. Lachance, Norma M. Morella, Neelendu Dey.

**Supervision:** Neelendu Dey.

**Validation:** Sean T. Koester, Naisi Li, Neelendu Dey.

**Visualization:** Sean T. Koester, Daniel M. Lachance, Norma M. Morella, Neelendu Dey.

**Writing – original draft:** Sean T. Koester, Naisi Li, Daniel M. Lachance, Norma M. Morella, Neelendu Dey.

**Writing – review & editing:** Sean T. Koester, Naisi Li, Daniel M. Lachance, Norma M. Morella, Neelendu Dey.

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
