## [Decision Letter · Decision Letter 0]

4 Feb 2021

PONE-D-20-36549

Respiratory and digestive tract variability in Ace2 expression associated with the microbiome

PLOS ONE

Dear Dr. Dey,

Thank you for submitting your manuscript to PLOS ONE. After careful consideration, we feel that it has merit but does not fully meet PLOS ONE’s publication criteria as it currently stands. Therefore, we invite you to submit a revised version of the manuscript that addresses the points raised during the review process.

We look forward to receiving your revised manuscript.

Kind regards,

Jane Foster, PhD

Academic Editor

PLOS ONE

Journal Requirements:

Reviewers' comments:

Reviewer's Responses to Questions

**Comments to the Author**

1. Is the manuscript technically sound, and do the data support the conclusions?

Reviewer #1: Yes

Reviewer #2: Yes

Reviewer #3: Yes

2. Has the statistical analysis been performed appropriately and rigorously? 

Reviewer #1: Yes

Reviewer #2: I Don't Know

Reviewer #3: No

3. Have the authors made all data underlying the findings in their manuscript fully available?

Reviewer #1: Yes

Reviewer #2: Yes

Reviewer #3: Yes

4. Is the manuscript presented in an intelligible fashion and written in standard English?

Reviewer #1: Yes

Reviewer #2: Yes

Reviewer #3: Yes

5. Review Comments to the Author

Reviewer #1: Koester et al. looked into the association between GI and respiratory tract microbiome and Ace2 expression in those sites. The manuscript is well written and clearly shows that microbiota can affect Ace2 expression in different tissues. My comments on this manuscript

Title

I would suggest the title could be Respiratory and …….is associated with microbiome

Introduction section

In paragraph 2, line 6 it is unclear what the author meant to say. Do they mean antibiotics treatment in germ free mice or SPF mice improve their phenotype? Please correct this accordingly.

Reviewer #2: Dear Authors,

This is a concise paper describing experiments completed assessing the expression of Ace2, primarily among mice, related to microbiome expression. It is interesting work that has potential to contribute to the body of work related to severity of COVID-19 infections, particularly among people with gut symptoms. From the data presented, it is difficult to know if increased expression of Ace2 in the gut would be associated with (worse) gut symptoms of those infected. Conceptually it is interesting and would be interesting to follow-up and see if increased expression is associated with increased risk and/or severity of COVID-19 infection. I have some relatively minor comment that I think would improve the paper. Mostly, I think the authors should be careful not to overstate results and add a little more detail to some of the methods and results. Thank you authors for your work and interesting paper.

I think respiratory should be removed from the title, as it seems only the gut results were significant.

The last statement in the introduction may be overstating the results, related to COVID-19 infection risk. Seems that the upregulation of expression of Ace2 may modulate severity, not necessarily risk for acquiring COVID-19. There is no data here that suggests a lower expression of Ace2 in the gut, since respiratory expression changes were modest, decreases risk of acquiring the disease. Recommend changing “infection risk and disease severity” to just disease severity, unless there is data that evaluates exposure to the virus and subsequent infection related to Ace2 expression.

After antibiotic depletion of the SPF mice, were the Ace2 expression levels the same as the GF mice or were they still lower than GF mice? Follow up question (if this wasn’t done, no need to add the experiment for this paper, but would be a cool follow-up study), did you look in a group of mice that were treated with antibiotics, but waited for the antibiotics to washout to see if the change in expression persists? Conversely, perhaps when the gut microbiota reverts back, Ace2 expression also reverts to pre-antibiotic levels? This could have interesting implications for individuals that had antibiotics prescribed early in the course of COVID-19 infection.

Regarding methods, it seems that the authors only counted genes annotated as peptidase or protease. Genes present don’t always associate with expression or protein levels – this should be addressed/made clear that this is what was correlated somewhere in the results/discussion.

Why were GF and SPF mice given different microbiome standard communities? This should be explained or at least noted in the main text, particularly since the GF mice had communities with a higher number of annotated genes. Was the fact they were given difference communities accounted for in the analysis? This could confound the results.

Given the genetic background of mice seems to be unequal for GF and SPF and SPF mice appear to be older than the GF mice, the authors should include the results that support these differences did not influence the results in the supplementary data file. The supplementary file just notes which mice were include in which paragraph and reads that each variable was analyzed separately. Since they are all together in one table, it’s really difficult to validate or decipher which animals go with which experiments/tests.

This first sentence in the discussion should include “in mice” at the end, since microbiome proteases/peditases did not evaluate Ace2 expression in human samples – only disease status, and there was no significant relationship reported.

In the conclusion paragraph, I think the authors should stop after strategies for downregulating Ace2, which may decrease susceptibility. There is no data presented that suggests upregulating Ace2 would facilitate COVID-19 recovery.

In supplementary table 2, the total Metagenome-encoded proteases/peptidases for communities 1 and 2 are reversed (i.e., the total for 2 is the sum of genes for community 1 and vice versa)

Reviewer #3: The authors have performed a straightforward and important study to determine whether microbial factors regulate ACE2 expression at different tissues. They used germ-free, gnotobiotic, and specific pathogen free (SPF) mice in the presence and absence of antibiotic perturbation of the host microbiome. ACE2 expression was measured via qPCR. Their results demonstrate an association between microbial colonization and ACE2 expression in the GI tract, and suggest that perturbation to a healthy community in the gut may increase ACE2 gene expression. They observed an interesting significant correlation between peptidase/protease-encoding bacterial taxa and ACE2 expression in the gut, which may relate to the role of ACE2 in amino acid transport. Overall, this is a significant and interesting study, and I have few minor comments.

Comments.

The description of methods are scant and would benefit from more detail in reference to the FMT, metagenome, and statistical analysis. Depending on the distribution of the annotated metagenome data, one-way ANOVA may not be appropriate.

It’s not clear where the taxa within each synthetic community originate (from ATCC, but unclear if they originate from human isolates), and whether the whole genome analysis confirming peptidase-encoding genes is from the isolates, sequenced directly, or from related bacteria from a public database.

That GF mice have higher ACE2 expression is puzzling and touched on briefly in the discussion. This merits further discussion, including perhaps being regulated by cytokines that may be aberrant in GF mice due to a lack of microbiota (so, ACE2 expression as in indirect effect of microbiome depletion/perturbation). This was partially addressed using il10-deficient mice.

One interesting finding was that a perturbation to the microbiome seemed to increase ACE2 expression, at least in the gut. Some discussion on whether diseases associated with microbiota perturbations may strengthen the impact of this manuscript.

6. PLOS authors have the option to publish the peer review history of their article (what does this mean?). If published, this will include your full peer review and any attached files.

Reviewer #1: **Yes: **Saroj Khatiwada

Reviewer #2: No

Reviewer #3: No

---

## [Author Response · Author response to Decision Letter 0]

2 Mar 2021

March 1, 2021

Jane Foster, PhD

Academic Editor

PLOS ONE

Dear Dr. Foster and colleagues, 

We thank you for your thoughtful reviews and for the opportunity to improve our manuscript (PONE-D-20-36549). Our responses to the very helpful comments are below, organized by reviewer and addressed in the order of the original comments: 

Reviewer #1

1) I would suggest the title could be Respiratory and …….is associated with microbiome

In order to address both this comment and comment #1 from Reviewer 2, we changed the title to, “Variability in digestive and respiratory tract Ace2 expression is associated with the microbiome.” 

2) In paragraph 2, line 6 it is unclear what the author meant to say. Do they mean antibiotics treatment in germ free mice or SPF mice improve their phenotype? Please correct this accordingly. 

We agree with this reviewer’s comment and modified this line in order to clarify and disambiguate: “Intriguingly, antibiotic-mediated microbiota depletion in SPF mice partially rescued the phenotype seen in GF mice, suggesting a capacity for microbiome-mediated regulation of Ace2 expression.”

Reviewer #2

1) I think respiratory should be removed from the title, as it seems only the gut results were significant.

In order to address both this comment and comment #1 from Reviewer 1, we changed the title to, “Variability in digestive and respiratory tract Ace2 expression is associated with the microbiome.” As we do not claim significance of respiratory Ace2 expression variation in the title, we hope that by leading with the word “variability,” the extent of our claims is clearer. Since the trend of higher respiratory tract Ace2 expression in germ-free mice was consistent and reproducible across multiple experiments, we feel that it is justified to include “respiratory” in the title; however, if the Editor feels that we should remove “respiratory” or edit the title otherwise, we would be happy to oblige.

2) The last statement in the introduction may be overstating the results, related to COVID-19 infection risk. Seems that the upregulation of expression of Ace2 may modulate severity, not necessarily risk for acquiring COVID-19. There is no data here that suggests a lower expression of Ace2 in the gut, since respiratory expression changes were modest, decreases risk of acquiring the disease. Recommend changing "infection risk and disease severity" to just disease severity, unless there is data that evaluates exposure to the virus and subsequent infection related to Ace2 expression.

We agree and removed “infection risk” from the final line of the introduction.

3) After antibiotic depletion of the SPF mice, were the Ace2 expression levels the same as the GF mice or were they still lower than GF mice? Follow up question (if this wasn't done, no need to add the experiment for this paper, but would be a cool follow-up study), did you look in a group of mice that were treated with antibiotics, but waited for the antibiotics to washout to see if the change in expression persists? Conversely, perhaps when the gut microbiota reverts back, Ace2 expression also reverts to pre-antibiotic levels? This could have interesting implications for individuals that had antibiotics prescribed early in the course of COVID-19 infection.

This is a really interesting comment. To answer the question, Ace2 expression levels were lower in antibiotics-treated SPF mice than in GF mice. We modified the end of the first paragraph of Results: “Intriguingly, SPF mice subjected to microbiota depletion via antibiotic administration for 10 days had higher intestinal Ace2 expression compared to mice that did not receive antibiotics (n=5-7/cohort, Fig 1B). Nonetheless, Ace2 transcript levels were not as high after antibiotics as in the GF state (p=0.08 and p=0.04 in proximal and distal colon, respectively, two-tailed Student’s t-test). This partial recapitulation of the GF phenotype in the gut, together with the absence of a measurable effect of antibiotics in the respiratory tract, suggests that antibiotic use may not have sizable immediate effects on host Ace2 expression.”

We agree the suggested follow-up studies would be quite interesting and perhaps expand the clinical implications of our study. We touch on this point in more general terms through the addition of the following sentence to the final paragraph of the Discussion: “Understanding the kinetics and magnitudes of these effects may have implications for antibiotic (and probiotic) use.”

4) Regarding methods, it seems that the authors only counted genes annotated as peptidase or protease. Genes present don't always associate with expression or protein levels – this should be addressed/made clear that this is what was correlated somewhere in the results/discussion.

We agree with this comment. We have edited the text in two areas: 

- To the penultimate paragraph of the Discussion, we added, “However, a caveat to this analysis is that core metabolic pathways are relatively evenly encoded in the human gut microbiome [17]; therefore, any variation in expressed proteases and peptidases that might influence host Ace2 expression is not captured in our analysis.” 

- We added the phrase “regulators of gene expression” to the second sentence of the third paragraph of Discussion, in which we describe confounders to our inability to correlate microbiome-encoded peptidases/proteases with Ace2 expression. 

5) Why were GF and SPF mice given different microbiome standard communities? This should be explained or at least noted in the main text, particularly since the GF mice had communities with a higher number of annotated genes. Was the fact they were given difference communities accounted for in the analysis? This could confound the results.

We had initially only stated in Methods, “Gnotobiotic mouse tissue samples were harvested as part of prior experiments unrelated to the present study.” To clarify this point within the main text, we added the following sentence to the second paragraph of the Results section to address this comment: “Tissues harvested as part of prior gnotobiotic experiments enabled us to assess whether Ace2 expression varies in different host and environmental contexts.”

6) Given the genetic background of mice seems to be unequal for GF and SPF and SPF mice appear to be older than the GF mice, the authors should include the results that support these differences did not influence the results in the supplementary data file. The supplementary file just notes which mice were include in which paragraph and reads that each variable was analyzed separately. Since they are all together in one table, it's really difficult to validate or decipher which animals go with which experiments/tests.

We address the point that variables such as age did not influence results in the following sentence in the lone paragraph of “Host and dietary factors” (final sub-section of the Results section): “Several variables were not significant determinants of gut or lung Ace2 expression: genetic background (C57BL/6 versus Swiss-Webster mice); Ret (gene critical to enteric nervous system development); Apc (tumor suppressor commonly mutated in colorectal cancer; Figure 2D); gender; age (comparing 9-16-week-old SPF wild-type mice); or a high-fat diet (p>0.05, one-way ANOVA tests performed for each sample tissue type for each variable; specific comparisons denoted in Table S1).” 

In Table S1, we re-organized the data as follows: first all wild-type mice, then mutant mice grouped by genotype; within the wild-type mice, we present GF, then SPF, then GF colonized with different microbial communities (grouped by each microbiota), then SPF mice treated with antibiotics, and finally SPF mice treated with antibiotics and re-colonized with defined communities. “MICE INCLUDED IN STATISTICAL CALCULATIONS PERFORMED IN THIS REPORT, STRATIFIED BY SPECIFIC COMPARISON” columns show an “X” in the row of each mouse included in a given comparison. 

7) This first sentence in the discussion should include "in mice" at the end, since microbiome proteases/peditases [sic] did not evaluate Ace2 expression in human samples – only disease status, and there was no significant relationship reported.

We agree and added “in mice” to the end of the first sentence in the discussion.

8) In the conclusion paragraph, I think the authors should stop after strategies for downregulating Ace2, which may decrease susceptibility. There is no data presented that suggests upregulating Ace2 would facilitate COVID-19 recovery.

We have modified the final line to read as follows: “Understanding these links could explain variability in COVID-19 severity and motivate strategies for modulating Ace2 in order to decrease susceptibility to infection (e.g. upregulating Ace2 in the elderly, in whom it is suspected that ACE2 levels become dangerously low [20]).” The issue of whether downregulation or upregulation is beneficial may be age-dependent (as shown below in a Figure from AlGhatrif and colleagues (reference 20) that we have copied and pasted here, in which it is proposed that critically low ACE2 levels can render elderly individuals to more severe COVID-19 infection. As such, upregulating Ace2 expression may be helpful in elderly populations. 

9) In supplementary table 2, the total Metagenome-encoded proteases/peptidases for communities 1 and 2 are reversed (i.e., the total for 2 is the sum of genes for community 1 and vice versa)

Thank you for astutely catching this error. Following review of our original analysis, we determined that during preparation of Table S2, we inadvertently swapped the numbers of genome-encoded proteases/peptidases for bacterial strains in communities 1 & 2. (The total numbers for each community are correct.) This error has been fixed accordingly. 

Reviewer #3

1) The description of methods are scant and would benefit from more detail in reference to the FMT, metagenome, and statistical analysis. Depending on the distribution of the annotated metagenome data, one-way ANOVA may not be appropriate.

We have expanded upon the following sub-sections of the Methods section: 

- We have added details regarding microbiota transplantation to the “Animal husbandry” sub-section describing preparation of inoculum, delivery of oral gavages, and duration of gut bacterial colonization.

- We have included additional details regarding ddPCR, and we introduce a Supplemental Figure showing these data.

- We have added details regarding genomic analysis to the “Quantification of bacterial genome-encoded proteases and peptidases” sub-section (see comment #2).

- We have added information about the sequencing data and our read alignment strategy to the “Estimation of microbiome-encoded proteases and peptidases in a metagenomic dataset” sub-section.

- We have added a sub-section to the methods called, “Data analysis,” and we have ensured that all statistical tests and results are clearly stated in the main text. In our review, we identified one instance in which the test we used (ANOVA) was not ideal for the data being analyzed, and instead we performed the Kruskal-Wallis test (penultimate paragraph of Results, immediately preceding “Host and dietary factors”). 

2) It's not clear where the taxa within each synthetic community originate (from ATCC, but unclear if they originate from human isolates), and whether the whole genome analysis confirming peptidase-encoding genes is from the isolates, sequenced directly, or from related bacteria from a public database.

To clarify this point, we added the following lines to the “Quantification of bacterial genome-encoded proteases and peptidases” sub-section of Methods: “All bacterial species used to assemble synthetic consortia were purchased from their respective vendors (American Type Culture Collection (ATCC) or the German Collection of Microorganisms and Cell Culture (DSM)); draft genomes of these type-strains are publicly available via NCBI. The most-recent genome assemblies of all bacteria in synthetic communities were downloaded from the NCBI Prokaryotic RefSeq database and annotated using Prokka (version 1.14.5) [8]. Gene annotations that included “peptidase” or “protease” were counted, with sums represented in Table S2.”

3) That GF mice have higher ACE2 expression is puzzling and touched on briefly in the discussion. This merits further discussion, including perhaps being regulated by cytokines that may be aberrant in GF mice due to a lack of microbiota (so, ACE2 expression as in indirect effect of microbiome depletion/perturbation). This was partially addressed using il10-deficient mice.

We agree with this reviewer that GF mice having higher levels of Ace2 is intriguing and added the following lines to the second paragraph of the discussion to address this. “One possible explanation is cytokines regulate the expression of Ace2. Furthermore, higher intestinal Ace2 mRNA levels among Il10 mutant mice supports this hypothesis. In this scenario, Ace2 expression is an indirect effect of microbial depletion.”

4) One interesting finding was that a perturbation to the microbiome seemed to increase ACE2 expression, at least in the gut. Some discussion on whether diseases associated with microbiota perturbations may strengthen the impact of this manuscript.

Whether differences in COVID-19 severity in individuals with diseases characterized by alterations in the gut microbiota (e.g. inflammatory bowel disease, obesity, metabolic syndrome, etc) are attributable to the gut microbiota is unclear. We hesitate to include this line in the Discussion, however, as it is highly speculative. 

We again thank you and the reviewers for their thorough and very helpful critiques of the paper. In addition to addressing these comments, we have reformatted the manuscript to conform to PLOS ONE style requirements.

Sincerely,

Neelendu Dey

---

## [Editor Report · Decision Letter 1]

4 Mar 2021

Variability in digestive and respiratory tract Ace2 expression is associated with the microbiome

PONE-D-20-36549R1

Dear Dr. Dey,

We’re pleased to inform you that your manuscript has been judged scientifically suitable for publication and will be formally accepted for publication once it meets all outstanding technical requirements.

Kind regards,

Jane Foster, PhD

Academic Editor

PLOS ONE

---

## [Editor Report · Acceptance letter]

8 Mar 2021

PONE-D-20-36549R1 

Variability in digestive and respiratory tract Ace2 expression is associated with the microbiome 

Dear Dr. Dey:

I'm pleased to inform you that your manuscript has been deemed suitable for publication in PLOS ONE. Congratulations! Your manuscript is now with our production department. 

Kind regards, 

on behalf of

Dr. Jane Foster 

Academic Editor

PLOS ONE